# ADAPTIVE BLOCK-WISE LEARNING FOR KNOWLEDGE DISTILLATION

## ABSTRACT

Knowledge distillation allows the student network to improve its performance under the supervision of transferred knowledge. Existing knowledge distillation methods are implemented under the implicit hypothesis that knowledge from teacher and student contributes to each layer of the student network to the same extent. In this work, we argue that there should be different contributions of knowledge from the teacher and the student during training for each layer. Experimental results evidence this argument. To the end, we propose a novel Adaptive Block-wise Learning (ABL) for Knowledge Distillation to automatically balance teacher-guided knowledge between self-knowledge in each block. Specifically, to solve the problem that the error backpropagation algorithm cannot assign weights to each block of the student network independently, we leverage the local error signals to approximate the global error signals on student objectives. Moreover, we utilize a set of meta variables to control the contribution of the student knowledge and teacher knowledge to each block during the training process. Finally, the extensive experiments prove the effectiveness of our method. Meanwhile, ABL provides an insightful view that in the shallow blocks, the weight of teacher guidance is greater, while in the deep blocks, student knowledge has more influence.

## 1 INTRODUCTION

Knowledge distillation (KD) in deep learning imitates the pattern of human learning. Hinton et al. (2015) proposes the original concept of KD, which minimizes the KL divergence between the logits of teacher (soft label) and student. This allows KD to be expressed as a mode in which a complex pre-trained model is used as a teacher to guide a lightweight student model learning. Following such teacher-student framework, a series of KD methods are mainly developed in the following directions: what, where and how to distill. No matter what kind of directions, these existing KD methods are based on the same implicit hypothesis that in the training process of distillation, the contribution of student's and teacher's knowledge to each layer of the student network is fixed, whether it is the last layer or the first layer. This is because after the error backpropagation (BP) (Rumelhart et al., 1986), the weight of the error signals on each layer is determined by the same hyper-parameters, which is shown in Figure 1(a). Intuitively, this limits the flexibility of balancing the knowledge of teacher and student, which harms excavating the potentialities of the student model.

Therefore, we argue that in the representation learning process of student network guided by teacher knowledge, different layers of the student network have different emphases on the knowledge learned through the one-hot labels and the knowledge distilled by the teacher. This means that some levels are more inclined to utilize student knowledge to learn, while others tend to leverage teacher knowledge. Furthermore, we also argue that the contribution of student and teacher knowledge to representation learning should be adaptive at each level. However, the existing KD methods obtain the global error signal from the last layer, which is hard to allocate the hierarchical weights.

To explore the student network hierarchically in the training process, we modify the backward computation graphs and leverage the *local error signals* by the family of local objectives (Jaderberg et al., 2017; Nøkland & Eidnes, 2019; Belilovsky et al., 2020; Pyeon et al., 2021) to approximate the *global error signals* generated by the last layer. These local loss functions focus on the local error signals and decoupled learning. Here, by leveraging auxiliary networks, we adopt the above local strategies to acquire the approximation of the global error signal created by the student loss

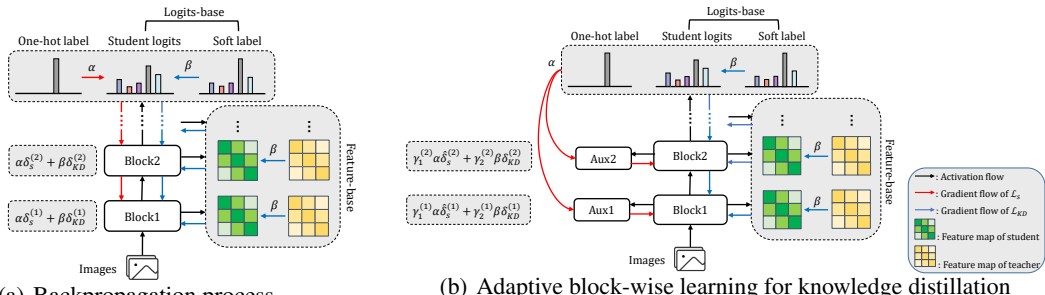

Figure 1: Comparison between the Backpropagation process of KD and adaptive block-wise learning for KD, from the perspective of gradient. The distilled knowledge can be based on logits or based on features. **(a)** The contributions of the knowledge from student and teacher are fixed and equal for different blocks. **(b)** The gradient flows are different for different blocks, and can be adaptively modified by meta variables $\gamma$.

objective corresponding to the one-hot labels. This provides the possibility to independently assign different weights to teacher knowledge and student knowledge at different layers.

After allowing the student error signals of each layer to obtain independently, the current issue is on exploring the balance between student knowledge and teacher knowledge at each layer. We model this issue as a bilevel optimization problem (Anandalingam & Friesz, 1992) by adding a set of meta variables on the error signals corresponding to two types of knowledge. These meta variables represent the influence of which target knowledge is preferred by the update of the corresponding layer. In addition, after the bilevel optimization based on the gradient descent solution, we obtain the optimal meta variables of the target network under the target KD method and utilize them for the final evaluation.

To the end, we propose a novel paradigm dubbed **A**daptive **B**lock-wise **L**earning (**ABL**) for Knowledge Distillation, which allows conventional teacher-student architecture to explore the influence of knowledge from teacher and student in blocks. As shown in Figure 1(b), the proposed method changes the error signals acquisition path of student objective function from global to local and adds a group of meta variables to measure the contribution of knowledge from students and teachers. Furthermore, we acquire the balance between knowledge from the student and the teacher on the validation set. Besides, we leverage the optimized meta variables to train the corresponding distillation method. Our main contributions are as follows:

1. We propose a novel paradigm named adaptive block-wise learning for knowledge distillation, which automatically balances the contribution of knowledge from the student and teacher for each block.

2. We discover that the deep and abstract representation inclines to learn from student knowledge, while the shallow and less abstract representation tends to be guided by teacher knowledge. We hope this discovery could provide another learning view for KD.

3. We conduct extensive experiments under eleven recent distillation benchmarks. Experimental results demonstrate the effectiveness of the proposed framework in improving the performance of existing distillation methods.

## 2 RELATED WORK

**Knowledge distillation.** Knowledge distillation usually transfers knowledge from large models to small models, which is completed under the teacher-student framework. The vanilla KD is firstly proposed by Hinton et al. (2015), which allows the student model to mimic the final prediction of the teacher model. Zhao et al. (2022) decouple the knowledge (binary probabilities) by the target knowledge and non-target knowledge. In addition to the above two logits-based KD, there are

also many methods based on features (Romero et al., 2015; Komodakis & Zagoruyko, 2017; Tung & Mori, 2019; Peng et al., 2019; Park et al., 2019; Ahn et al., 2019; Tian et al., 2020), which are mainly aimed at transferring the knowledge of intermediate features. In addition to the above consideration of what kind of knowledge to transfer, Song et al. (2022) considers where to transfer knowledge, Mirzadeh et al. (2020), Son et al. (2021) and Chen et al. (2021) explore how to transfer knowledge. However, none of these focuses on an important question, *i.e.* how to balance the transferred knowledge and the knowledge from the student model, which is more directly reflected in the effect of the target student model.

**Local error signals.** In the field of decoupled neural networks, many researches are looking for an efficient way to replace the global error signals with local error signals. DNI (Jaderberg et al., 2017) utilizes the linear layer for each layer to learn synthetic gradient by approximating the gradient of backpropagation. Nøkland & Eidnes (2019) obtains the local error signals with similarity matching and prediction local loss to unlock the backward updating. DGL (Belilovsky et al., 2020) designs the more efficient auxiliary networks MLP-SR to create the local error. SEDONA (Pyeon et al., 2021) adds the neural architecture search technique into greedy learning to obtain the more appropriate local signals meliorated by the global signals. Following these work, we approximate the global error signals of student model with the local error signals.

## 3 METHODS

We begin by describing the knowledge distillation from the perspective of gradient (Section 3.1). We then introduce our proposed adaptive block-wise learning for knowledge distillation (Section 3.2), where the global error signals are approximated with the local signals to realize a changeable weight allocation of the knowledge from teacher and student. Finally, we optimize simultaneously the global error and the local error signals by adopting a bilevel optimization (Section 3.3).

### 3.1 KNOWLEDGE DISTILLATION FROM THE PERSPECTIVE OF GRADIENT

Existing knowledge distillation methods can be divided into two parts: 1) One part is to learn by oneself through the student, 2) and the other is to learn by distilling the teacher's knowledge, which can be expressed as:

$$\mathcal{L} = \alpha\mathcal{L}_S + \beta\mathcal{L}_{KD}, \tag{1}$$

where $\mathcal{L}_S$ is usually cross-entropy loss (De Boer et al., 2005) between the predicted probabilities from only the student model and the label, $\mathcal{L}_{KD}$ can be the Kullback-Leibler divergence (Hershey & Olsen, 2007) in vanilla knowledge distillation, $\ell_2$-norm distance for the intermediate representations in Fitnets (Romero et al., 2015), or any feature-based distillation loses. $\alpha$ and $\beta$ balance the contribution of $\mathcal{L}_S$ and $\mathcal{L}_{KD}$.

With the above analysis, we rethink knowledge distillation from the perspective of gradient during the training of the student model. We now consider student networks consisting of $L$ layers $f^{(l)}, l \in \{1, 2, \ldots, L\}$, each outputting $h^{(l)}$ through the parameters $\theta^{(l)}$. Thus, the updating of the parameters with the gradient-based update rule can be formulated as:

$$\theta^{(l)} \leftarrow \theta^{(l)} - \eta\frac{\partial\mathcal{L}}{\partial h^{(l)}}\frac{\partial h^{(l)}}{\partial\theta^{(l)}}, \tag{2}$$

where $\eta$ represents the learning rate and $\dfrac{\partial\mathcal{L}}{\partial h^{(l)}}$ is the backpropagated error gradients, which is usually represented by $\delta^{(l)}$. By Eq. (1), the training process of KD which reflects in the error gradients of each layer in the student network can be written as:

$$\delta^{(l)} = \alpha\delta_S^{(l)} + \beta\delta_{KD}^{(l)}, \tag{3}$$

where $\delta_S^{(l)}$ and $\delta_{KD}^{(l)}$ denote the contributions of the student's knowledge and teacher's knowledge in the $l$-th layer of the student network, respectively.

To facilitate our statements, we define $\delta_S^{(l)}$ and $\delta_{KD}^{(l)}$ as **student error signals** and **teacher error signals** respectively. Intuitively, no matter how $\alpha$ and $\beta$ are set, the error gradients of each layer can be expressed by Eq. (3). This means that the contribution allocation of the two parts of each layer is fixed, which is shown in Figure 1(a).

---

**Algorithm 1** Bilevel optimization for Adaptive Block-wise Learning for Knowledge Distillation

---

Initialize meta variables $\gamma$ as 0
Warm-start input network weights $(\theta, \varphi)$ with Eq. (7) and Eq. (8)
 1: **while** not converged **do**
 2:     Calculate $(\theta^*, \varphi^*)$ on training mini-batch with Eq. (7) and Eq. (8)
 3:     Updata $\gamma$ on validation mini-batch with Eq. (9)
 4:     Updata $(\theta, \varphi)$ on training mini-batch with Eq. (7) and Eq. (8)
 5: **end while**
Obtain weight allocation variables $\gamma$ of the global teacher error and the local student error

---

### 3.2 APPROXIMATING WITH LOCAL ERROR SIGNALS

Our goal is to realize the adaptive block-wise weight allocation of the knowledge from the student model and the teacher model in KD. However, it is impossible to assign weights in blocks by using the error backpropagation algorithm in conventional teacher-student frameworks, which is shown in Eq. (3). Thus, we make an approximate estimation of the global error signal of the student objective generated by backpropagation, leveraging the block-wise local error signal. Based on the transformation of this gradient flow, our main task is equivalent to the following two steps: (1) generating a local gradient flow that approximates the gradient flow created by the last layer, and (2) simultaneously optimizing the gradient error signals generated by the local gradient flow and by the backward propagation, achieving adaptive weight assignment for each block. These two steps will be explored in Section 3.2 and Section 3.3.

For step (1), we imitate supervised greedy learning to assign the local error signals which are used to approximate the error obtained by backpropagation. Following previous works (Nøkland & Eidnes, 2019; Belilovsky et al., 2020; Pyeon et al., 2021), we utilize the auxiliary networks and target vectors to create the student error (local error signals) independently of the blocks, which is shown in Figure 1(b). We usually identify a single block as an input network consisting of convolutional layers, normalization and pooling layers. As for the input networks with residual connections, we regard each residual connection as a block. Each block corresponds to an auxiliary network to predict the target and compute the local objective. Meanwhile, this method is used for obtaining the local student error of each block. Note that, for the last block, the auxiliary network is the last fully-connected (FC) layer of the student.

Let $X^{(l)}$ and $Y$ denote the output representation for block $l$ and the labels, $l \in \{0, 1, \ldots, L-1\}$. $X^{(0)}$ means the input data. $\varphi^{(l)}$ represents the parameters of the auxiliary networks corresponding to the $l$-th block. We define the local objective function as $\hat{\mathcal{L}}^{(l)}(\theta^{(l)}, \varphi^{(l)}; X^{(l)}, Y)$. To apply our paradigm to the training process of the existing KD framework as shown in Eq. (3), we maintain both sensitive hyper-parameters $\alpha$ and $\beta$. In addition, we propose to adopt a set of meta variables $\gamma^{(l)} \in \mathbb{R}^2$ in a two-dimensional continuous domain. It is used to modify the contribution of student error and teacher error to the total error of different network layers. To be specific, we utilize a softmax over all possible values, namely, $\gamma_1^{(l)}, \gamma_2^{(l)} = \mathrm{softmax}(\gamma^{(l)})$, as the weight allocation variables. Finally, we can formulate the loss objective function for block $l$ as follows:

$$\mathcal{L}^{(l)}(\theta, \varphi, \gamma; X^{(0)}, Y, Y^T) = \gamma_1^{(l)} \alpha \hat{\mathcal{L}}^{(l)}(\theta^{(l)}, \varphi^{(l)}; X^{(l)}, Y) + \gamma_2^{(l)} \beta \mathcal{L}_{KD}^{(l)}(\theta^{(l)}; X^{(0)}, Y^T), \quad (4)$$

where $Y^T$ is the distilled signals of the teacher model. Finally, the error gradient of non-last block $l$ can be re-defined as:

$$\hat{\delta}^{(l)} = \gamma_1^{(l)} \alpha \hat{\delta}_S^{(l)} + \gamma_2^{(l)} \beta \delta_{KD}^{(l)}, \quad (5)$$

where the local student error $\hat{\delta}_S^{(l)} = \dfrac{\partial \hat{\mathcal{L}}^{(l)}}{\partial h^{(l)}}$ and the teacher error $\delta_{KD}^{(l)} = \dfrac{\partial \mathcal{L}_{KD}^{(l)}}{\partial h^{(l)}}$. As for the last block, we use $\hat{\delta}^{(L)} = \dfrac{\partial \mathcal{L}}{\partial h^{(L)}}$.

### 3.3 OPTIMIZING META VARIABLES WITH BILEVEL OPTIMIZATION

As mentioned in Section 3.2, we need to optimize simultaneously both the global teacher error and the local student error to ensure a competitive performance of the student network. The objective

is noted by Eq. (4) and Eq. (5), which is influenced not only by the network parameters $(\theta, \varphi)$ but also by the meta variables $\gamma$. This implies that this optimization problem can be solved as a bilevel optimization problem (Anandalingam & Friesz, 1992; Liu et al., 2019; Pyeon et al., 2021) with $(\theta, \varphi)$ as the inner-loop variables and $\gamma$ as the outer-loop variables, which can be defined as:

$$\min_{\gamma} \mathcal{L}_{val}(\theta^*, \varphi^*, \gamma) \qquad s.t. \ (\theta^*, \varphi^*) = \arg\min_{\theta, \varphi} \mathcal{L}_{train}(\theta, \varphi, \gamma), \tag{6}$$

where $\mathcal{L}_{train}$ and $\mathcal{L}_{val}$ denote the loss function on training and validation set. By Eq. (6), we adopt the nested method which uses the training set to update the model parameters $(\theta, \varphi)$ with global error signals and local error signals of Eq. (5) and the validation set to optimize the meta parameters $\gamma$. Specifically, we fix $\gamma$ and update $(\theta, \varphi)$ with a mini-batch training data in the inner loop. The error gradients updating can be written as:

$$\theta^{(l)}(\gamma) \leftarrow \theta^{(l)}(\gamma) - \eta \, \hat{\delta}^{(l)}(\gamma) \frac{\partial h^{(l)}}{\partial \theta^{(l)}}, \tag{7}$$

$$\varphi^{(l)}(\gamma) \leftarrow \varphi^{(l)}(\gamma) - \eta \, \nabla_{\varphi} \hat{\mathcal{L}}_{train}(\theta^{(l)}, \varphi^{(l)}), \tag{8}$$

where $\hat{\mathcal{L}}_{train}$ is the local objective of the training set and $l \in \{0, 1, \ldots, L\}$. In the outer loop, we optimize the meta variables $\gamma$ with a mini-batch of validation set based on $(\theta, \varphi)$ updated by Eq. (7) and Eq. (8) as follows:

$$\gamma \leftarrow \gamma - \lambda \, \nabla_{\gamma} \mathcal{L}_{val}((\theta, \varphi) - \eta \nabla_{\theta, \varphi} \mathcal{L}_{train}(\theta, \varphi, \gamma), \gamma), \tag{9}$$

where $\lambda$ is the learning rate of the meta variable $\gamma$. Then we apply the chain rule and the finite difference approximation (Liu et al., 2019) to the last item of Eq. (9):

$$\nabla_{\gamma} \mathcal{L}_{val}(\cdot, \gamma) \approx \nabla_{\gamma} \mathcal{L}_{val}(\theta^*, \varphi^*, \gamma) - \eta \frac{\nabla_{\gamma} \mathcal{L}_{train}(\theta^+, \varphi^+, \gamma) - \nabla_{\gamma} \mathcal{L}_{train}(\theta^-, \varphi^-, \gamma)}{2\xi}, \tag{10}$$

where $\mathcal{L}_{val}$ is the validation loss, $(\theta^*, \varphi^*)$ is the updated results of Eq. (7) and Eq. (8), $(\theta^{\pm}, \varphi^{\pm}) = (\theta, \varphi) \pm \xi \nabla_{(\theta^*, \varphi^*)} \mathcal{L}_{val}(\theta^*, \varphi^*, \gamma)$, $\xi$ is a scale [1].

Finally, we fix the reasonable $\gamma$ to optimize the network parameters $(\theta, \varphi)$ on the training set and use the updated network parameters as the initial value of the next iteration. The bilevel optimization training process is shown in Algorithm 1.

**Warm start** We adopt the warm start technique (Pyeon et al., 2021) on the network weights $(\theta, \varphi)$ with fixed meta variables $\gamma$ ($\gamma = 0$), which makes bilevel optimization stably and avoids the bad results of $\gamma$ caused by inappropriate initialization of $(\theta, \varphi)$. In our experiments, we pre-train the student with 40000 iterations to obtain a satisfactory initial value for the optimization of $\gamma$.

**Two-stage training** Our proposed method adopts a two-stage training strategy, including meta variables optimization and final evaluation. According to the optimized meta variables, we evaluate the network weights $(\theta, \varphi)$ with the training loss to obtain the final comparable results, by following Eq. (7) and Eq. (8).

## 4 EXPERIMENTS

We first describe the main experimental settings (Section 4.1) which are necessary to understand our work. Then, we provide the results of baseline comparisons on several datasets (Section 4.2). Finally, we construct ablation experiments (Section 4.3) and some further discussions (Section 4.4) to explore the necessity and effectiveness of the components of our framework.

### 4.1 EXPERIMENTAL SETTINGS

**Datasets.** **CIFAR-100** (Krizhevsky et al., 2009) is a image classification dataset which consists of 60000 $32\times32$ colour images 100 classes. **ImageNet** (Deng et al., 2009) is a large-scale classification dataset with 1.2 million $256\times256$ images from 1000 classes. As for the setting of dataset split, we divide the official training set into our training set and validation set, which are 1:1 in size. This is to satisfy the conditions of bilevel optimization of meta variables. However, our warm start and final evaluation are still under the official setting.

---

[1]The same setting with Liu et al. (2019), $\xi = 0.01/||\nabla_{(\theta^*, \varphi^*)} \mathcal{L}_{val}(\theta^*, \varphi^*, \gamma)||_2$

| | | ResNet56 72.35 | | | ResNet110 74.31 | | | ResNet32×4 79.42 | | | WRN-40-2 75.61 | | | VGG13 74.64 | | |
| --- | --- | --- | --- | --- | --- | --- | --- | --- | --- | --- | --- | --- | --- | --- | --- | --- |
| teacher | | ResNet20 69.06 | | | ResNet32 71.14 | | | ResNet8×4 72.50 | | | WRN-40-1 73.26 | | | VGG8 70.36 | | |
| student | | Stan. | Adap. | Δ | Stan. | Adap. | Δ | Stan. | Adap. | Δ | Stan. | Adap. | Δ | Stan. | Adap. | Δ |
| **logits** | KD | 70.66 | 71.01 | +0.35 | 73.08 | 73.85 | +0.77 | 73.33 | 73.57 | +0.24 | 73.54 | 74.14 | +0.60 | 72.98 | 73.95 | +0.97 |
| | DKD | 71.97 | 72.23 | +0.26 | 74.11 | 74.25 | +0.14 | 76.32 | 76.15 | -0.17 | 74.81 | 75.07 | +0.26 | 74.68 | 75.37 | +0.69 |
| **features** | Fitnets | 69.21 | 69.60 | +0.39 | 71.06 | 71.20 | +0.14 | 73.50 | 73.70 | +0.20 | 72.24 | 72.54 | +0.30 | 71.02 | 71.74 | +0.72 |
| | AT | 70.55 | 71.04 | +0.49 | 72.31 | 73.20 | +0.89 | 73.44 | 74.38 | +0.94 | 72.77 | 73.53 | +0.76 | 71.43 | 73.13 | +0.70 |
| | SP | 69.67 | 71.06 | +1.39 | 72.69 | 73.07 | +0.38 | 72.94 | 74.20 | +1.26 | 72.43 | 73.27 | +0.84 | 72.68 | 73.52 | +0.84 |
| | CC | 69.63 | 69.91 | +0.28 | 71.48 | 72.19 | +0.71 | 72.97 | 73.46 | +0.49 | 72.21 | 72.04 | -0.15 | 70.71 | 71.67 | +0.96 |
| | RKD | 69.61 | 69.83 | +0.21 | 71.82 | 72.16 | +0.34 | 71.90 | 73.22 | +1.32 | 72.22 | 72.45 | +0.23 | 71.48 | 71.69 | +0.21 |
| | VID | 70.38 | 70.75 | +0.37 | 72.61 | 72.85 | +0.24 | 73.09 | 73.59 | +0.40 | 73.30 | 72.96 | -0.34 | 71.23 | 71.74 | +0.51 |
| | PKT | 70.34 | 70.96 | +0.62 | 72.61 | 73.02 | +0.41 | 73.64 | 74.64 | +0.98 | 73.45 | 73.40 | -0.05 | 72.88 | 73.06 | +0.88 |
| | CRD | 71.16 | 71.40 | +0.24 | 73.48 | 74.09 | +0.61 | 75.51 | 75.69 | +0.18 | 74.14 | 74.25 | +0.11 | 73.94 | 73.95 | +0.01 |
| | ReviewKD | 71.89 | 71.94 | +0.05 | 73.89 | 74.27 | +0.38 | 75.63 | 75.86 | +0.23 | 75.09 | 75.33 | +0.21 | 74.84 | 74.97 | +0.13 |

Table 1: Test accuracy (%) of **homogeneous** distillation on the CIFAR-100. **Stan.**, **Adap.**, Δ denote the standard KD method, the standard KD method within our adaptive block-wise framework and the performance improvement over the corresponding standard KD method, respectively.

**Backbone and Auxiliary Networks**  We adopt several backbone architectures as our main networks, including ResNet (He et al., 2016), VGG (Simonyan & Zisserman, 2015), Wide ResNet (Zagoruyko & Komodakis, 2016), ShuffleNet (Zhang et al., 2018), and MobieNet (Howard et al., 2017). Moreover, the selection of auxiliary networks should not only utilize lightweight auxiliary networks, but also ensure the parallel training of the auxiliary networks and main networks. Thus, we build an auxiliary block with a point-wise convolutional layer, a depth-wise convolutional layer, an average pooling layer, and a fully-connected layer.

**Baselines**  We compare the performance of the existing KD benchmarks themselves with those that use ABL. We divide these baselines into two categories: logits-based and feature-based. Logits-based distillation methods include KD (Hinton et al., 2015) and DKD (Zhao et al., 2022). The feature-based distillation methods are Fitnets (Romero et al., 2015), AT (Komodakis & Zagoruyko, 2017), SP (Tung & Mori, 2019), CC (Peng et al., 2019), RKD (Park et al., 2019), VID (Ahn et al., 2019), PKT (Passalis & Tefas, 2018), CRD (Tian et al., 2020), ReviewKD Chen et al. (2021).

More details of these experimental settings are shown in Appendix A.1.

## 4.2   MAIN RESULTS

**Results on CIFAR-100**  We evaluate our framework on CIFAR-100. Table 1 and Table 2 show the performance of the standard KD baselines and them in our proposed framework. In Table 1, we adopt various student-teacher combinations in same-style architectures. Table 2 shows the results on heterogeneous distillation of several student-teacher frameworks.

For both types of distillation, we evaluate with five teacher-student combinations and eleven existing (including the state-of-the-art) knowledge distillation methods. After using our framework, despite no tuning for each task, the performance of most methods have improved by 0.5%-2%, which proves the effectiveness and wide applicability of our framework. In particular, AT, a feature-based method that plays a negative role in heterogeneous distillation, has a significant 8.1% improvement after our adaptive block-wise training. The extensive experimental results on both homogeneous and heterogeneous distillations demonstrate the effectiveness and portability of the proposed adaptive block-wise learning for KD.

**Results on ImageNet**  We also evaluate the performance of several methods in our proposed framework on the large-scale ImageNet dataset, which is shown in Table 3 and Table 4. These experimental methods include KD and DKD based on logits and AT and ReviewKD based on features. As can be seen from both tables, the performance of the four methods we selected has steadily improved on the accuracy of top-1 and top-5 by an average of 0.3-0.5% after using our adaptive framework. These results on ImageNet verify the scalability of our proposed method.

| | | VGG13 74.64 | | | ResNet50 79.34 | | | ResNet32×4 79.42 | | | ResNet32×4 79.42 | | | WRN-40-2 75.61 | | |
|---|---|---|---|---|---|---|---|---|---|---|---|---|---|---|---|---|
| teacher | | MobileNetV2 64.60 | | | MobileNetV2 64.60 | | | ShuffleNetV1 70.50 | | | ShuffleNetV2 71.28 | | | ShuffleNetV1 70.50 | | |
| student | | Stan. | Adap. | Δ | Stan. | Adap. | Δ | Stan. | Adap. | Δ | Stan. | Adap. | Δ | Stan. | Adap. | Δ |
| **logits** | KD | 67.37 | 68.85 | +1.48 | 67.35 | 68.81 | +1.46 | 74.07 | 75.36 | +1.29 | 74.45 | 76.22 | +1.77 | 74.83 | 75.69 | +0.86 |
| | DKD | 69.71 | 69.88 | +0.17 | 70.35 | 70.81 | +0.46 | 76.45 | 77.03 | +0.58 | 77.07 | 77.89 | +0.82 | 76.70 | 76.87 | +0.17 |
| **features** | Fitnets | 64.14 | 64.15 | +0.01 | 63.16 | 63.31 | +0.15 | 73.59 | 74.49 | +0.90 | 73.54 | 74.95 | +1.41 | 73.73 | 74.28 | +0.55 |
| | AT | 59.40 | 65.76 | +6.36 | 58.58 | 66.68 | +8.10 | 71.73 | 75.07 | +3.34 | 72.73 | 76.39 | +3.66 | 73.32 | 75.55 | +2.23 |
| | SP | 66.30 | 67.59 | +1.29 | 68.08 | 67.68 | -0.40 | 73.48 | 75.77 | +2.29 | 74.56 | 76.62 | +2.06 | 74.52 | 76.10 | +1.58 |
| | CC | 64.86 | 65.20 | +0.34 | 65.43 | 65.47 | +0.04 | 71.14 | 71.84 | +0.70 | 71.29 | 73.83 | +2.54 | 71.38 | 72.14 | +0.76 |
| | RKD | 64.52 | 64.97 | +0.45 | 64.43 | 65.40 | +0.97 | 72.28 | 73.01 | +0.73 | 73.21 | 74.86 | +0.65 | 72.21 | 74.28 | +2.07 |
| | VID | 65.56 | 66.42 | +0.86 | 67.57 | 65.87 | -1.70 | 73.38 | 73.01 | -0.37 | 73.40 | 73.68 | +0.28 | 73.61 | 74.44 | +0.83 |
| | PKT | 67.13 | 67.44 | +0.31 | 66.52 | 67.07 | +0.55 | 74.10 | 74.23 | +0.13 | 74.69 | 75.86 | +1.17 | 73.89 | 74.74 | +0.85 |
| | CRD | 69.73 | 69.88 | +0.15 | 69.11 | 70.02 | +0.91 | 75.11 | 75.71 | +0.60 | 75.65 | 76.83 | +1.18 | 76.05 | 76.50 | +0.45 |
| | ReviewKD | 70.37 | 70.68 | +0.31 | 69.89 | 70.70 | +0.81 | 77.45 | 77.98 | +0.53 | 77.78 | 77.94 | +0.16 | 77.14 | 77.41 | +0.25 |

Table 2: Test accuracy (%) of **heterogeneous** distillation on the CIFAR-100. **Stan.**, **Adap.**, Δ denote the standard KD method, the standard KD method within our adaptive block-wise framework and the performance improvement over the corresponding standard KD method, respectively.

| Setting | Teacher | Student | KD | | | DKD | | | AT | | | ReviewKD | | |
|---|---|---|---|---|---|---|---|---|---|---|---|---|---|---|
| | ResNet50 | MobileNetV2 | Stan. | Adap. | Δ | Stan. | Adap. | Δ | Stan. | Adap. | Δ | Stan. | Adap. | Δ |
| top-1 | 76.16 | 68.89 | 70.47 | 70.37 | -0.10 | 72.05 | 72.49 | +0.44 | 69.56 | 68.79 | -0.77 | 72.56 | 72.61 | +0.05 |
| top-5 | 92.86 | 88.76 | 89.46 | 89.98 | +0.52 | 91.05 | 91.84 | +0.79 | 89.33 | 89.51 | +0.18 | 91.00 | 92.15 | +1.15 |

Table 3: Test accuracy (%) of KD on the ImageNet between the **different**-style architecture.

| Setting | Teacher | Student | KD | | | DKD | | | AT | | | ReviewKD | | |
|---|---|---|---|---|---|---|---|---|---|---|---|---|---|---|
| | ResNet34 | ResNet18 | Stan. | Adap. | Δ | Stan. | Adap. | Δ | Stan. | Adap. | Δ | Stan. | Adap. | Δ |
| top-1 | 73.31 | 69.75 | 70.66 | 70.97 | +0.31 | 71.70 | 72.24 | +0.54 | 70.69 | 71.09 | +0.40 | 71.61 | 71.86 | +0.25 |
| top-5 | 91.42 | 89.07 | 89.88 | 90.01 | +0.13 | 90.41 | 90.95 | +0.44 | 90.01 | 90.37 | +0.36 | 90.51 | 90.62 | +0.11 |

Table 4: Test accuracy (%) of KD on the ImageNet between the **same**-style architecture.

## 4.3 ABLATION STUDIES

In this section, we conduct several ablation experiments to verify the effectiveness of our proposed framework. Specially, we explore the framework deeply by answering the following questions.

*1) Are the local error signals good approximations of the global error signals?* If the local error signals are not representative of the global ones, then there will be a significant gap between the final results and the comparison results. To explore the effectiveness of local error signals from the local loss objective, we set the value of meta variables as zero. This means that the effect on any parameters in the input neural network is the same as the student error and the teacher error, which is similar to the conventional backpropagation KD framework. We build the experiments with several distillation methods under both same-style and different-style architecture setting and the results on CIFAR-100 are shown in Table 5. From the results, for the KD, AT, RKD, ReviewKD method, the model updated with local error signals performs better, while in the DKD, CRD method, the model updated with global error signals performs better. Overall, the error between the two is less than 1%. Thus, most of the teacher-student architectures we compared can be updated with local error signals to achieve the performance of conventional training strategies.

*2) Are the adaptive meta variables reasonable?* To validate the effectiveness of meta variables used for balancing the student error and teacher error, we conduct three different strategies of ABL for KD via meta variables: *Fixed-distillation*, *Random-distillation* and *No-distillation*. Fixed-distillation is based on the assumption that the contribution of student's knowledge is the same as that of teacher's knowledge, and both the corresponding meta variables are 0.5. Random-distillation randomly decides the balance between knowledge from teacher and student.

| Teacher | ResNet110 | | ResNet32×4 | | VGG13 | | ResNet50 | | ResNet32×4 | |
|---|---|---|---|---|---|---|---|---|---|---|
| Student | ResNet32 | | ResNet8×4 | | VGG8 | | MobileNetV2 | | ShuffleNetV1 | |
| | Global | Local | Global | Local | Global | Local | Global | Local | Global | Local |
| KD | 73.08 | **73.28** | **73.33** | 73.13 | 72.98 | **73.93** | 67.35 | **68.74** | 74.07 | **75.14** |
| DKD | **74.11** | 73.62 | **76.32** | 76.02 | 74.68 | **74.82** | **70.35** | 69.31 | **76.45** | 75.73 |
| AT | 72.31 | **73.18** | 73.44 | **74.21** | 71.43 | **72.44** | 58.58 | **64.61** | 71.73 | **75.04** |
| RKD | 71.82 | **72.73** | 71.90 | **72.73** | 71.48 | **72.37** | 64.43 | **64.98** | 72.28 | **73.21** |
| CRD | **73.48** | 73.39 | 75.51 | **75.71** | **73.94** | 73.82 | **69.11** | 68.51 | **75.11** | 74.62 |
| ReviewKD | 73.89 | **74.06** | **75.63** | 75.28 | **74.84** | 73.93 | **69.89** | 66.32 | **77.45** | 77.15 |

Table 5: Ablation studies on local error signals. Test accuracy (%) of knowledge distillation on the CIFAR-100. **Global** and **Local** denote student error from the global signals and the local signals.

| Setting | ResNet110→ResNet32 | | | | VGG13→VGG8 | | | | ResNet50→MobileNetV2 | | | | ResNet32×4→ShuffleNetV1 | | | |
|---|---|---|---|---|---|---|---|---|---|---|---|---|---|---|---|---|
| | Adaptive | Fixed | Random | No | Adaptive | Fixed | Random | No | Adaptive | Fixed | Random | No | Adaptive | Fixed | Random | No |
| KD | **73.85** | 73.28 | 73.39 | 71.89 | **73.95** | 73.93 | 73.36 | 71.82 | **68.81** | 68.74 | 68.16 | 65.15 | **75.36** | 75.14 | 75.02 | 72.81 |
| DKD | **74.25** | 73.62 | 73.68 | 71.89 | **75.37** | 74.82 | 74.17 | 71.82 | **70.81** | 69.31 | 69.06 | 65.15 | **77.03** | 75.73 | 75.60 | 72.81 |
| AT | **73.20** | 73.18 | 72.50 | 71.89 | **73.13** | 72.44 | 72.71 | 71.82 | **66.68** | 64.61 | 64.15 | 65.15 | **75.07** | 75.04 | 74.83 | 72.81 |
| RKD | 72.16 | **72.73** | 71.67 | 71.89 | **71.69** | 71.37 | 71.15 | 71.82 | **65.40** | 64.98 | 64.03 | 65.15 | 73.01 | **73.21** | 72.88 | 72.81 |
| CRD | **73.48** | 73.39 | 73.03 | 71.89 | **73.94** | 73.82 | 73.43 | 71.82 | **69.11** | 68.51 | 67.23 | 65.15 | **75.11** | 74.62 | 75.01 | 72.81 |
| ReviewKD | **74.27** | 74.06 | 72.92 | 71.89 | **74.84** | 73.93 | 74.13 | 71.82 | **70.70** | 66.32 | 65.33 | 65.15 | **77.45** | 77.15 | 75.43 | 72.81 |

Table 6: Ablation studies on the block-wise distillation strategy. Test accuracy (%) of distillation methods on the CIFAR-100. **Adaptive**, **Fixed**, **Random** and **No** represent the adaptive-type, fixed-type, random-type and no-type distillation, respectively.

No-distillation means the student trains using ABL without the teacher's signal. Several experiments are conducted on CIFAR-100 under the setting of both homogeneous and heterogeneous distillation. The results are shown in Table 6, which demonstrates that the performance of our adaptive-distillation strategy has a consistent improvement above other comparable baselines.

We also visualize the final evaluation process corresponding to the meta variables with ResNet20 as student and ResNet56 as teacher

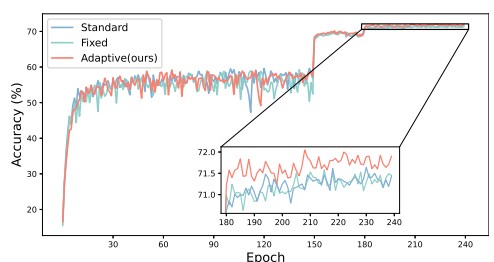

Figure 2: Comparison of learning curves

based on DKD in Figure 2. We compare the training curves of three strategies including Standard-type, Fixed-type and Adaptive-type distillation. These three processes are similar in the initial training stage, while in the final stage, our proposed ABL has a relatively stable improvement. This result shows the effectiveness of the adaptive-distillation strategy with reasonable meta variables.

## 4.4 FURTHER INVESTIGATION ON ADAPTIVE BLOCK-WISE LEARNING

To further analyze ABL, we deeply explore the optimization process of meta variables and the results from the block-wise views.

**Optimization process of meta variables**   We draw the meta variables training process in Figure 3(a) and 3(b). In these figures, with the increase in the number of network blocks, the network parameters prefer their own error signals (student error), while the influence of teacher error on total gradient error is reduced. In particular, in the shallow layers (the first three blocks), the parameter update of ResNet20 is more affected by teacher error signals. This means that from the perspective of gradient flow, the shallow representation of the student model is more likely to be guided by the distilled teacher knowledge, while the deeper and more abstract representation of the student model is more inclined to update from self-knowledge.

**Block-wise results**   The results in Figure 4 demonstrate shallow representation does not have the ability to classify with high accuracy and fit teacher's deep representation. In terms of the specific approach in our framework, DKD achieves better results than other methods in the accuracy of each block, which is also proved in Figure 4(a) and 4(b). The difference of the correlation matrices

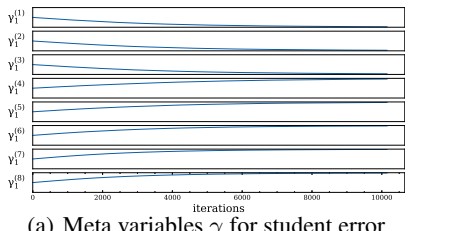 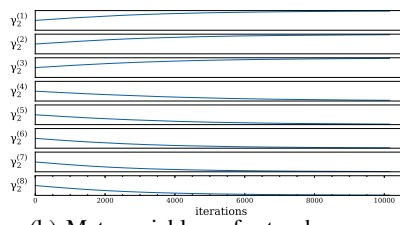

(a) Meta variables $\gamma$ for student error    (b) Meta variables $\gamma$ for teacher error

Figure 3: All experiments are conducted on CIFAR-100 based on DKD with ResNet20 as student and ResNet56 as teacher. Figure **(a)** and **(b)** show the meta variables $\gamma$ optimization process for student error and teacher error, respectively. The $x$-axis and $y$-axis represent the number of iterations and the softmax values (range from 0 to 1) of the meta variables, respectively. Note that the values of the meta variables are set to 0 at iteration 0, thus all the initial points in the figure are 0.5.

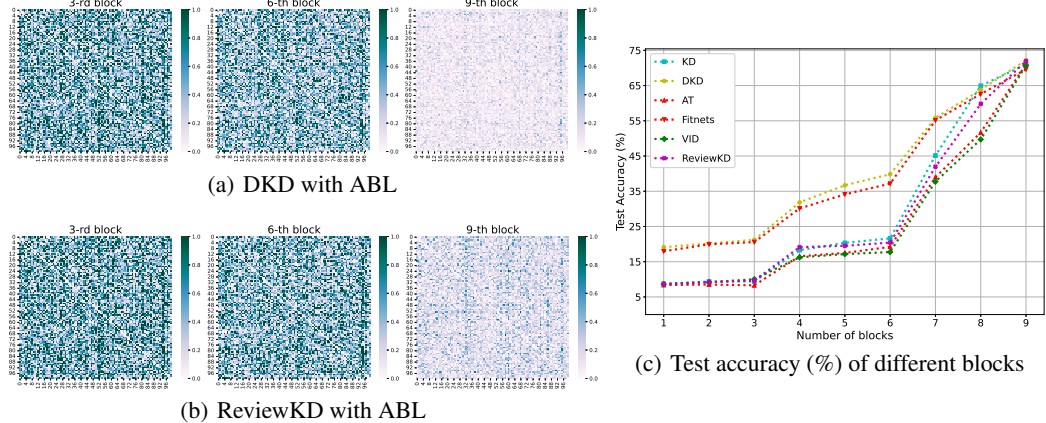

(a) DKD with ABL

(b) ReviewKD with ABL

(c) Test accuracy (%) of different blocks

Figure 4: All the experiments are implemented under the setting of ResNet20 as student ResNet56 as teacher. **(a)** and **(b)** represent the correlations matrices between class logits of student and teacher in different blocks. For ResNet20, a residual block consists of three blocks that we define. **(c)** draws the test accuracy of nine blocks of ResNet20 based on ABL for six types of distillation methods.

between the logits of the student and teacher after DKD distillation is less than ReviewKD. In Figure 4(c), the results show DKD with ABL achieves better performance on each block. Interestingly, although the test accuracy of Fitnets distillation method is worse than other results in overall results, it is in the lead in the first six blocks. The reason is that, in the experimental setting, the hint-layer in Fitnets method is set to the second residual block, *i.e.*, in the sixth block we define, which results in that only the first six blocks have teacher knowledge guidance. These observations present a block-wise perspective, which provide a more detailed view of the distillation procedures.

## 5 CONCLUSION

In this paper, we provide a novel viewpoint on knowledge distillation, which discusses the balance between the teacher's and student's knowledge at different levels. Existing distillation methods are built on an implicit hypothesis that teacher knowledge and student knowledge make the same contribution to the learning of shallow and deep representations. However, we consider that for shallow representations, the student network is easy to train under the guidance of teacher representation, while for deep representation, the student model is difficult to imitate directly from teacher's representation knowledge. Thus, we propose Adaptive Block-wise Learning for Knowledge Distillation, which leverages a set of meta variables to control the balance between the student's local error signals and the teacher's global error signals. The experimental results prove the effectiveness of the proposed method. More importantly, we confirm the hypothesis that the guidance of the teacher's knowledge to the student network does decrease with the increase of blocks.

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

## A APPENDIX

### A.1 IMPLEMENTATION DETAILS

We train our models on CIFAR-100 for 240 epochs with SGD that the weight decay is 5e-4 and the momentum is 0.9. Due to the batch size being 64, the initial learning rate is set to 0.01 for ShuffleNet and MobieNet and 0.05 for others, and the learning rate is divided by 10 at 150, 180 and 210 epochs. For ImageNet, we train the models for epochs 100. We set the batch size to 128, and initialize the learning rate to 0.1, which is divided by 10 at 30, 60, and 90 epochs.

| Method | KD(Hinton et al., 2015) | DKD(Zhao et al., 2022) | Fitnets(Romero et al., 2015) | AT(Komodakis & Zagoruyko, 2017) |
|---|---|---|---|---|
| $\beta$ | 0.9 | 1.0 | 100 | 1000 |
| Method | SP(Tung & Mori, 2019) | CC(Peng et al., 2019) | RKD(Park et al., 2019) | VID(Ahn et al., 2019) |
| $\beta$ | 3000 | 0.02 | 1.0 | 1.0 |
| Method | PKT(Passalis & Tefas, 2018) | CRD(Tian et al., 2020) | ReviewKDChen et al. (2021) | |
| $\beta$ | 30000 | 0.8 | 8.0 | |

Table 7: Setting of distillation loss factor $\beta$.

By following Zhao et al. (2022), Chen et al. (2021) and Chen et al. (2021), we train the student model with the following objective:

$$\mathcal{L} = \alpha\mathcal{L}_{CE} + \beta\mathcal{L}_{KD}, \tag{11}$$

For the setting of the hyper-parameters $\alpha$, all methods are 1 except that KD is 0.1. The setting of distillation loss factor $\beta$ is shown in Table 7. Specially, for the logits-base distillation methods including KD and DKD, we set the temperature $T = 4$.

All experiments are conducted with 2 NVIDIA TESLA V100S GPU cards.

## A.2 STANDARD DEVIATION FOR RESULTS ON CIFAR-100

We conduct our all experiments on CIFAR-100 over 5 trials. In Table 8 and Table 9, we provide the standard deviation of 5 trials on 11 benchmarks.

| | | ResNet56 72.35 | | ResNet110 74.31 | | ResNet32×4 79.42 | | WRN-40-2 75.61 | | VGG13 74.64 | |
| | | ResNet20 69.06 | | ResNet32 71.14 | | ResNet8×4 72.50 | | WRN-40-1 73.26 | | VGG8 70.36 | |
| teacher/student | | Stan. | Adap. | Stan. | Adap. | Stan. | Adap. | Stan. | Adap. | Stan. | Adap. |
|---|---|---|---|---|---|---|---|---|---|---|---|
| logits | KD | $70.66_{\pm0.24}$ | $\mathbf{71.01_{\pm0.06}}$ | $73.08_{\pm0.18}$ | $\mathbf{73.85_{\pm0.14}}$ | $73.33_{\pm0.25}$ | $\mathbf{73.57_{\pm0.12}}$ | $73.54_{\pm0.20}$ | $\mathbf{74.14_{\pm0.28}}$ | $72.98_{\pm0.19}$ | $\mathbf{73.95_{\pm0.12}}$ |
| | DKD | $71.97_{\pm0.41}$ | $\mathbf{72.23_{\pm0.18}}$ | $74.11_{\pm0.21}$ | $\mathbf{74.25_{\pm0.38}}$ | $\mathbf{76.32_{\pm0.02}}$ | $76.15_{\pm0.03}$ | $74.81_{\pm0.08}$ | $\mathbf{75.07_{\pm0.13}}$ | $74.68_{\pm0.32}$ | $\mathbf{75.37_{\pm0.29}}$ |
| features | Fitnets | $69.21_{\pm0.36}$ | $\mathbf{69.60_{\pm0.12}}$ | $71.06_{\pm0.13}$ | $\mathbf{71.20_{\pm0.44}}$ | $73.50_{\pm0.28}$ | $\mathbf{73.70_{\pm0.18}}$ | $72.24_{\pm0.24}$ | $\mathbf{72.54_{\pm0.27}}$ | $71.02_{\pm0.31}$ | $\mathbf{71.74_{\pm0.03}}$ |
| | AT | $70.55_{\pm0.27}$ | $\mathbf{71.04_{\pm0.45}}$ | $72.31_{\pm0.08}$ | $\mathbf{73.20_{\pm0.01}}$ | $73.44_{\pm0.19}$ | $\mathbf{74.38_{\pm0.29}}$ | $72.77_{\pm0.10}$ | $\mathbf{73.53_{\pm0.25}}$ | $71.43_{\pm0.09}$ | $\mathbf{73.13_{\pm0.28}}$ |
| | SP | $69.67_{\pm0.20}$ | $\mathbf{71.06_{\pm0.42}}$ | $72.69_{\pm0.41}$ | $\mathbf{73.07_{\pm0.05}}$ | $72.94_{\pm0.23}$ | $\mathbf{74.20_{\pm0.22}}$ | $72.43_{\pm0.27}$ | $\mathbf{73.27_{\pm0.01}}$ | $72.68_{\pm0.19}$ | $\mathbf{73.52_{\pm0.45}}$ |
| | CC | $69.63_{\pm0.32}$ | $\mathbf{69.91_{\pm0.19}}$ | $71.48_{\pm0.21}$ | $\mathbf{72.19_{\pm0.18}}$ | $72.97_{\pm0.17}$ | $\mathbf{73.46_{\pm0.45}}$ | $\mathbf{72.21_{\pm0.25}}$ | $72.04_{\pm0.41}$ | $70.71_{\pm0.24}$ | $\mathbf{71.67_{\pm0.27}}$ |
| | RKD | $69.61_{\pm0.06}$ | $\mathbf{69.83_{\pm0.46}}$ | $71.82_{\pm0.34}$ | $\mathbf{72.16_{\pm0.36}}$ | $71.90_{\pm0.11}$ | $\mathbf{73.22_{\pm0.09}}$ | $72.22_{\pm0.20}$ | $\mathbf{72.45_{\pm0.34}}$ | $71.48_{\pm0.05}$ | $\mathbf{71.69_{\pm0.23}}$ |
| | VID | $70.38_{\pm0.14}$ | $\mathbf{70.75_{\pm0.23}}$ | $72.61_{\pm0.17}$ | $\mathbf{72.85_{\pm0.43}}$ | $73.09_{\pm0.21}$ | $\mathbf{73.59_{\pm0.28}}$ | $\mathbf{73.30_{\pm0.13}}$ | $72.96_{\pm0.45}$ | $71.23_{\pm0.23}$ | $\mathbf{71.74_{\pm0.05}}$ |
| | PKT | $70.34_{\pm0.04}$ | $\mathbf{70.96_{\pm0.21}}$ | $72.61_{\pm0.17}$ | $\mathbf{73.02_{\pm0.19}}$ | $73.64_{\pm0.18}$ | $\mathbf{74.64_{\pm0.37}}$ | $\mathbf{73.45_{\pm0.19}}$ | $73.40_{\pm0.30}$ | $72.88_{\pm0.09}$ | $\mathbf{73.06_{\pm0.07}}$ |
| | CRD | $71.16_{\pm0.17}$ | $\mathbf{71.40_{\pm0.13}}$ | $73.48_{\pm0.13}$ | $\mathbf{74.09_{\pm0.33}}$ | $75.51_{\pm0.18}$ | $\mathbf{75.69_{\pm0.11}}$ | $74.14_{\pm0.22}$ | $\mathbf{74.25_{\pm0.16}}$ | $\mathbf{73.94_{\pm0.22}}$ | $\mathbf{73.95_{\pm0.13}}$ |
| | ReviewKD | $71.89_{\pm0.31}$ | $\mathbf{71.94_{\pm0.33}}$ | $73.89_{\pm0.17}$ | $\mathbf{74.27_{\pm0.29}}$ | $75.63_{\pm0.15}$ | $\mathbf{75.86_{\pm0.31}}$ | $75.09_{\pm0.18}$ | $\mathbf{75.33_{\pm0.41}}$ | $74.84_{\pm0.14}$ | $\mathbf{74.97_{\pm0.08}}$ |

Table 8: Test accuracy (%) of homogeneous distillation on the CIFAR-100. Stan., Adap. denote the standard KD and the standard KD within our adaptive block-wise framework respectively.

| | | VGG13 74.64 | | ResNet50 79.34 | | ResNet32×4 79.42 | | ResNet32×4 79.42 | | WRN-40-2 75.61 | |
| | | MobileNetV2 64.60 | | MobileNetV2 64.60 | | ShuffleNetV1 70.50 | | ShuffleNetV2 71.28 | | ShuffleNetV1 70.50 | |
| teacher/student | | Stan. | Adap. | Stan. | Adap. | Stan. | Adap. | Stan. | Adap. | Stan. | Adap. |
|---|---|---|---|---|---|---|---|---|---|---|---|
| logits | KD | $67.37_{\pm0.32}$ | $\mathbf{68.85_{\pm0.21}}$ | $67.35_{\pm0.32}$ | $\mathbf{68.81_{\pm0.01}}$ | $74.07_{\pm0.19}$ | $\mathbf{75.36_{\pm0.27}}$ | $74.45_{\pm0.27}$ | $\mathbf{76.22_{\pm0.04}}$ | $74.83_{\pm0.17}$ | $\mathbf{75.69_{\pm0.22}}$ |
| | DKD | $69.71_{\pm0.08}$ | $\mathbf{69.88_{\pm0.46}}$ | $70.35_{\pm0.17}$ | $\mathbf{70.81_{\pm0.16}}$ | $76.45_{\pm0.09}$ | $\mathbf{77.03_{\pm0.20}}$ | $77.07_{\pm0.28}$ | $\mathbf{77.89_{\pm0.16}}$ | $76.70_{\pm0.11}$ | $\mathbf{76.87_{\pm0.11}}$ |
| features | Fitnets | $64.14_{\pm0.50}$ | $\mathbf{64.15_{\pm0.41}}$ | $63.16_{\pm0.47}$ | $\mathbf{63.31_{\pm0.26}}$ | $73.59_{\pm0.15}$ | $\mathbf{74.49_{\pm0.25}}$ | $73.54_{\pm0.22}$ | $\mathbf{74.95_{\pm0.42}}$ | $73.73_{\pm0.32}$ | $\mathbf{74.28_{\pm0.26}}$ |
| | AT | $59.40_{\pm0.20}$ | $\mathbf{65.76_{\pm0.28}}$ | $58.58_{\pm0.54}$ | $\mathbf{66.68_{\pm0.29}}$ | $71.73_{\pm0.31}$ | $\mathbf{75.07_{\pm0.08}}$ | $72.73_{\pm0.31}$ | $\mathbf{76.39_{\pm0.17}}$ | $73.32_{\pm0.35}$ | $\mathbf{75.55_{\pm0.20}}$ |
| | SP | $66.30_{\pm0.38}$ | $\mathbf{67.59_{\pm0.14}}$ | $\mathbf{68.08_{\pm0.38}}$ | $67.68_{\pm0.23}$ | $73.48_{\pm0.42}$ | $\mathbf{75.77_{\pm0.21}}$ | $74.56_{\pm0.22}$ | $\mathbf{76.62_{\pm0.24}}$ | $74.52_{\pm0.24}$ | $\mathbf{76.10_{\pm0.34}}$ |
| | CC | $64.86_{\pm0.25}$ | $\mathbf{65.20_{\pm0.41}}$ | $65.43_{\pm0.15}$ | $\mathbf{65.47_{\pm0.27}}$ | $71.14_{\pm0.06}$ | $\mathbf{71.84_{\pm0.32}}$ | $71.29_{\pm0.38}$ | $\mathbf{73.83_{\pm0.22}}$ | $71.38_{\pm0.25}$ | $\mathbf{72.14_{\pm0.34}}$ |
| | RKD | $64.52_{\pm0.45}$ | $\mathbf{64.97_{\pm0.17}}$ | $64.43_{\pm0.42}$ | $\mathbf{65.40_{\pm0.02}}$ | $72.28_{\pm0.39}$ | $\mathbf{73.01_{\pm0.38}}$ | $73.21_{\pm0.28}$ | $\mathbf{74.86_{\pm0.10}}$ | $72.21_{\pm0.16}$ | $\mathbf{74.28_{\pm0.34}}$ |
| | VID | $65.56_{\pm0.42}$ | $\mathbf{66.42_{\pm0.06}}$ | $\mathbf{67.57_{\pm0.28}}$ | $65.87_{\pm0.16}$ | $\mathbf{73.38_{\pm0.09}}$ | $73.01_{\pm0.30}$ | $73.40_{\pm0.17}$ | $\mathbf{73.68_{\pm0.29}}$ | $73.61_{\pm0.12}$ | $\mathbf{74.44_{\pm0.36}}$ |
| | PKT | $67.13_{\pm0.30}$ | $\mathbf{67.44_{\pm0.37}}$ | $66.52_{\pm0.33}$ | $\mathbf{67.07_{\pm0.27}}$ | $74.10_{\pm0.25}$ | $\mathbf{74.23_{\pm0.17}}$ | $74.69_{\pm0.34}$ | $\mathbf{75.86_{\pm0.21}}$ | $73.89_{\pm0.16}$ | $\mathbf{74.74_{\pm0.37}}$ |
| | CRD | $69.73_{\pm0.42}$ | $\mathbf{69.88_{\pm0.13}}$ | $69.11_{\pm0.28}$ | $\mathbf{70.02_{\pm0.31}}$ | $75.11_{\pm0.32}$ | $\mathbf{75.71_{\pm0.33}}$ | $75.65_{\pm0.10}$ | $\mathbf{76.83_{\pm0.13}}$ | $76.05_{\pm0.14}$ | $\mathbf{76.50_{\pm0.24}}$ |
| | ReviewKD | $70.37_{\pm0.34}$ | $\mathbf{70.68_{\pm0.14}}$ | $69.89_{\pm0.45}$ | $\mathbf{70.70_{\pm0.23}}$ | $77.45_{\pm0.26}$ | $\mathbf{77.98_{\pm0.15}}$ | $77.78_{\pm0.21}$ | $\mathbf{77.94_{\pm0.19}}$ | $77.14_{\pm0.19}$ | $\mathbf{77.41_{\pm0.27}}$ |

Table 9: Test accuracy (%) of heterogeneous distillation on the CIFAR-100.

## A.3 MORE ABLATION STUDIES

*1) Is there a problem of gradient vanishing?* Gradient vanishing usually exists in networks with too many layers. To verify whether such problem exists in the standard KD scheme, if there is a

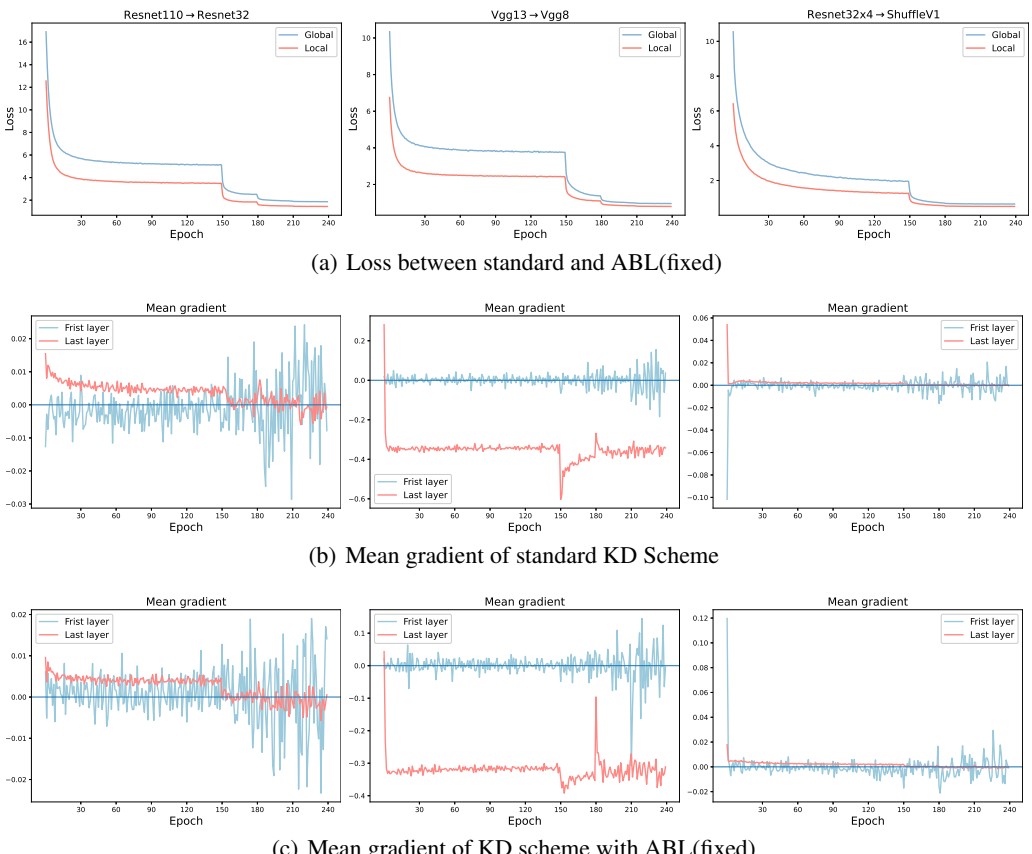

(a) Loss between standard and ABL(fixed)

(b) Mean gradient of standard KD Scheme

(c) Mean gradient of KD scheme with ABL(fixed)

Figure 5: All the experiments are implemented under the same KD setting. The first column takes ResNet32 as the student and ResNet110 as the teacher. The second column takes Vgg8 as the student, Vgg13 as the teacher. The third column sets ShuffleV1 as the student and ResNet32x4 as the teacher. **(a)** represents the learning curves between the standard and ABL. **(b)** figures the mean gradient of standard KD scheme. **(c)** draws the mean gradient of the KD scheme with ABL.

| Setting | ResNet110→ResNet32 | | | | VGG13→VGG8 | | | | ResNet50→MobileNetV2 | | | | ResNet32×4→ShuffleNetV1 | | | |
|---|---|---|---|---|---|---|---|---|---|---|---|---|---|---|---|---|
| | Adap. | Stan. | Stan.(tea.) | Adap.(tea.) | Adap. | Stan. | Stan.(tea.) | Adap.(tea.) | Adap. | Stan. | Stan.(tea.) | Adap.(tea.) | Adap. | Stan. | Stan.(tea.) | Adap.(tea.) |
| KD | **73.85** | 73.08 | 73.62 | 73.34 | **73.95** | 72.98 | 73.48 | 73.92 | **68.81** | 67.35 | 68.64 | 68.55 | **75.36** | 74.07 | 74.46 | 74.94 |
| DKD | **74.25** | 74.11 | 73.55 | 73.16 | **75.37** | 74.68 | 74.07 | 74.14 | **70.81** | 70.35 | 70.49 | 70.26 | **77.03** | 76.45 | 76.73 | 75.70 |

Table 10: Ablation studies on the block-wise distillation strategy. Test accuracy (%) of distillation methods on the CIFAR-100. **Adap.**, **Stan.**, **Stan.(tea.)** and **Adap.(tea.)** represent the adaptive-type, standard-type, standard(only teacher)-type and fix(only teacher)-type distillation, respectively.

problem, whether proposed local objectives has been improved by solving the problem, we figure the training process of the standard KD framework and our proposed KD framework, as well as the mean value of the gradient of the first and last layers under the same setting, which is shown in Figure 5. The results of Figure 5(a) show that the training process of the standard KD framework and our proposed KD framework is very stable. The loss is always changing, which proves that neither of the two frameworks can be trained because gradient vanishing. In Figure 5(b) and Figure 5(c), there is no difference of magnitude between the mean gradient of the first layer and the last layer, and there is no stagnant training. This proves that there is no gradient vanishing in both our ABL scheme and the standard KD scheme. The more important thing is the similarity of the both schemes images verify that local error signals are good approximations of the global error signals.

*2) Are the adaptive meta variables effective?* We construct more experiments to verify the effective of our proposed adaptive meta variables, which shown in Table 10. We create two additional

strategies: only using teacher knowledge as a standard KD with supervision (*Stan.(tea.)*), and the proposed ABL strategy under only teacher guidance (*Adap.(tea.)*). In the Table 10, the standard KD scheme utilized teacher knowledge only performs better than that supervised by both teacher knowledge and the ground truth knowledge. This proves that standard KD framework limits the potential of teacher knowledge. However, the performance of ABL framework with only teacher knowledge is poorer than that under both teacher's and student's knowledge. This shows that the meta variables effectively help the model to learn under the guidance of both student knowledge and teacher knowledge.

*3) How to design an appropriate auxiliary network?* The only goal of the auxiliary network in ABL is to achieve the possibility of local error instead of global error to create a later inquiry into the meta weight allocation of student knowledge contribution and teacher knowledge contribution. Thus, the performance of an appropriate auxiliary network need to approach to standard KD scheme (with the only supervision of last layer). This means that the auxiliary network should not be too simple (shallow) to achieve the effect of using local error instead of global error, nor should it be so complex as to greatly increase the difficulty and duration of training.

Inspired by the greedy learning (Jaderberg et al., 2017; Nøkland & Eidnes, 2019; Belilovsky et al., 2020; Pyeon et al., 2021), we conduct several types of auxiliary network as follow.

*Aux1*: AvgPool + FC layer,

*Aux2*: AvgPool + 3 point-wise convolutional layers + AvgPool + 3-layer MLP,

*Aux3*: a point-wise convolutional layer + a depth-wise convolutional layer + AvgPool + FC layer,

*Aux4*: a point-wise convolutional layer + a depth-wise convolutional layer + a inverted residual block + AvgPool + FC layer

The Table 11 figures the results on testing accuracy (%) and the training time (GPU hours).

| Setting | ResNet110→ResNet32 | | | | | VGG13→VGG8 | | | | |
|---|---|---|---|---|---|---|---|---|---|---|
| | Global | Aux1 | Aux2 | Aux3 | Aux4 | Global | Aux1 | Aux2 | Aux3 | Aux4 |
| Testing accuracy (%) | 73.08 | 71.83 | 72.97 | 73.28 | 73.63 | 72.98 | 72.61 | 74.01 | 73.93 | 74.33 |
| Training time (GPU hours) | 3.42 | 3.04 | 5.31 | 4.52 | 5.95 | 1.92 | 1.54 | 3.21 | 2.62 | 3.59 |

Table 11: Ablation studies on the auxiliary networks. All experiments are conducted under the same KD distillation setting.

### A.4 TRAINING EFFICIENCY

Due to the existence of auxiliary networks for computing local error and bi-level optimization for meta variables, ABL needs more training costs. The results of training efficiency are shown in Table 12. The total training time of ABL is about 1.25-1.35 times of that of the standard KD, but the performance of ABL improves by 0.5%-2% over most distillation methods, which proves these costs are desirable.

| Setting(GPU hours) | Bi-level optimization | Final evaluation | Total(ours) | Standard |
|---|---|---|---|---|
| MobileNetV2 | 1.04 | 4.81 | 5.85 | 4.65 |
| VGG8 | 0.57 | 2.05 | 2.62 | 1.92 |
| ResNet32 | 0.71 | 3.81 | 4.52 | 3.42 |
| ShuffleV1 | 1.07 | 4.98 | 6.05 | 4.67 |

Table 12: Training costs(GPU hours) of different models evaluate under KD distillation methods on the CIFAR-100. The total training process contains a bi-level optimization and a final evaluation.

### A.5 MORE INVESTIGATION ON ADAPTIVE BLOCK-WISE LEARNING

To explore the two types of knowledge in different blocks, we plot the test accuracy of each block of different student networks under the same DKD distillation setting, which is shown in Figure

6. In the Figure 6, the results demonstrate that each block of the student networks trained by the proposed ABL scheme outperforms that under the fixed ABL setting. In the shallow blocks, the teachers knowledge contributes more than student knowledge, and the accuracy is higher than the fixed one. While in the deep blocks, the student knowledge contributes more, and the accuracy is higher than the fixed one. Thus, teacher knowledge is more suitable to guide shallow representation, while student knowledge is more appropriate to guide deep representation.

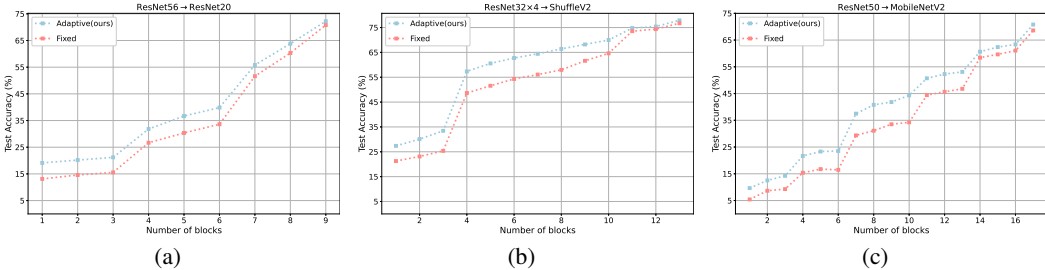

Figure 6: All the experiments are implemented under the same DKD setting. Fixed represents the model trained with the fixed contribution of student's knowledge and teacher's knowledge. Adaptive means the model under ABL training.

