# OpenReview forum: "Adaptive Block-wise Learning for Knowledge Distillation"
_ICLR.cc/2023/Conference — Submitted to ICLR 2023_

### Official Review · Reviewer_Tp5S · 2022-10-24

**Confidence:** 4
**Correctness:** 3
**Technical Novelty And Significance:** 2
**Empirical Novelty And Significance:** Not applicable
**Recommendation:** 3

**Clarity, Quality, Novelty And Reproducibility:**

The basic ideas of this paper are easy to understand, but the presentation is not good enough. This paper has serious issues in novelty, experiments and claims. Code is not provided.

Please refer to my comments in 'Strength And Weaknesses' for details.

**Strength And Weaknesses:**

Strengths.

+ Making the knowledge from teacher to layers of the student network to be different is interesting.

+ The proposed ABL sometimes shows improvements (but actually are not fair) to different KD methods.

+ Experimental comparisons are conducted on both CIFAR100 and ImageNet datasets with different teacher-student network pairs.

Weaknesses.

- The method.

The core ideas of the proposed ABL are in two aspects: 1) adding auxiliary networks to some intermediate student layers to ease the training; 2) in optimization, auxiliary networks are used to approximate the global error signals from the pre-trained teacher, learning to dynamically generate meta weighting factors for gradients propagation at different blocks of the student network. However, there already exist many KD works that explore the use of auxiliary networks to improve KD process. Some representative works are DKS [1], BYOT [2], DCM [3] , MetaDistiller [4], to name a few. Besides, MetaDistiller and some other works also explore the use of meta learning. Unfortunately, these works are completely missed by the authors. A comprehensive comparison of ABL with them, both in methodology and performance is necessary.

[1] Deeply-Supervised Knowledge Synergy, CVPR 2019.

[2] Be Your Own Teacher: Improve the Performance of Convolutional Neural Networks via Self Distillation, ICCV 2019.

[3] Knowledge Transfer via Dense Cross-layer Mutual-distillation, ECCV 2020.

[4] MetaDistiller: Network Self-Boosting via Meta-Learned Top-Down Distillation, ECCV 2020.

- The motivation.

In the paper, the authors argue that different layers of the student network have different emphases on the knowledge learned through the one-hot labels and the knowledge distilled by the teacher. Furthermore, the authors claim a new contribution that the deep and abstract representation inclines to learn from student knowledge, while the shallow and less abstract representation tends to be guided by teacher knowledge. However, the analysis and experiments provided in the paper are not convincing enough. Could the authors provide more evidence or richer experiments to support these claims? E.g., how about the role of auxiliary networks? how to design them? the effects of  different auxiliary classifiers? how about the performance of adding low-quality/shallow auxiliary networks? what will happen if smoothing the predication of the pre-trained teacher model?

- The experiments.

Experimental comparisons are problematic and misleading: (1) For experimental comparisons (Table 1 and Table 2) on CIFAR100, it seems that all results for counterpart baseline methods (denoted as 'Stan.') are directly copied from the paper of CRD (mostly), and the paper of DKD. As a result, the authors merely tested the combination of ABL and each of them, but not tested each corresponding counterpart with the same training machine and training code settings. As a result, all reported \delta values are totally misleading; (2) For experimental comparisons (Table 3 and Table 4) on ImageNet, the authors also use such an unfair comparison; (3) This even applies to some ablations. In a nutshell, the authors did not actually run any experiments for counterpart baseline methods (denoted as 'Stan.') at all. Therefore, main experiments need to re-design and re-implement. Even w.r.t. the current results, the improvement from ABL is mostly marginal.

A comprehensive comparison of ABL with closely related methods such as DKS [1], BYOT [2], DCM [3] and MetaDistiller [4] is necessary.

As auxiliary classifiers play a key role in ABL, a deep analysis of them is also necessary, please see my comments in 'The motivation' for details.

How about the training cost of ABL compared to counterpart baselines methods?

**----Update----**

I keep my original score as my major concerns are not well addressed.

**Summary Of The Paper:**

This paper attempts to improve knowledge distillation (KD) research from the perspective of allocating adaptive weighting factors to update layer-wise learnable parameters of the student model during back-propagation. The authors conjecture that the knowledge from the teacher  should be different to the student layers from shallow to deep. Based on this, they propose a new KD method called adaptive block-wise learning (ABL), which uses auxiliary networks for learning to dynamically generate meta weighting factors for gradients propagation at different blocks of the student network. In designs, these auxiliary networks are added to intermediate student layers, and leverage the local error signals to approximate the global error signals on student objectives. In implementation, ABL uses a greedy bi-level (two-stage) optimization. The effectiveness of ABL is validated on image classification datasets CIFAR100 and ImageNet.

**Summary Of The Review:**

This paper is below the acceptance bar of ICLR.

Please refer to my comments in 'Strength And Weaknesses' for details.

---

### Official Review · Reviewer_XFGM · 2022-10-25

**Confidence:** 2
**Correctness:** 3
**Technical Novelty And Significance:** 3
**Empirical Novelty And Significance:** 3
**Recommendation:** 8

**Clarity, Quality, Novelty And Reproducibility:**

The paper is clearly written and appears technically sound. The quality of the proposed method and the experimental results is high. Although local error signals have been studied in the past, the proposed approach seems novel, although I am not fully familiar with knowledge distillation literature. The reproducibility is enough for a researcher in the area.

**Strength And Weaknesses:**

Strengths:
- The paper presents a novel idea of using local errors to determine the contribution of teacher and student networks during knowledge distillation.
- The paper is well written and well organised.
- The paper reports relevant findings for the knowledge distillation area, in particular that deep blocks of the network benefit more from student knowledge, while shallow blocks of the network benefit from more teacher knowledge.

Weaknesses:
- Some terms are used that are never defined. This makes the paper difficult to understand for a researcher not in this area. For example, what are homogeneous and heterogeneous knowledge distillation? Please provide a definition or cite a relevant definition.
- In the experiments, it is never mentioned how big is the teacher and how big is the student for each of the datasets. Furthermore, how relevant is this for the proposed method?
- I would like to see how the findings regarding the contribution of teacher and student at different layers in the knowledge distillation setting are connected to previous similar findings in transfer learning in general (see for example [1])

References:
[1] Neyshabur, B., Sedghi, H., & Zhang, C. (2020). What is being transferred in transfer learning?. Advances in neural information processing systems, 33, 512-523.

**Summary Of The Paper:**

This paper proposes Adaptive Block-wise Learning, a method that adapts the amount of knowledge distillation from a teacher to a student at each layer of the network. A set of auxiliary networks is used for this purpose. The method uses local error signals to control how much teacher or student knowledge should each layer use. This is accompanied by a set of metavariables to control these contributions, which are optimized using bilevel optimization. The authors report experimental results on several SOTA knowledge distillation methods, showing that this more selective knowledge distillation strategy can benefit both homogeneous and heterogeneous knowledge distillation problems.

**Summary Of The Review:**

The paper presents a method for knowledge distillation that considers the contribution of teacher and students distinctly at different layers of the network. This approach seems novel and useful for the area. The conclusions around the different degrees of contribution of teacher and students at different depths of the network seem to be an important contribution (although I am not fully aware of the knowledge distillation literature, so I may be convinced otherwise based on other reviews). Based on this, my recommendation at this stage is for acceptance of the paper.

---

### Official Review · Reviewer_Jk2C · 2022-10-29

**Confidence:** 3
**Correctness:** 3
**Technical Novelty And Significance:** 3
**Empirical Novelty And Significance:** 3
**Recommendation:** 5

**Clarity, Quality, Novelty And Reproducibility:**

There is minimal novelty and no technical depth to the paper.

+ There is no description of the teacher, student or auxiliary network.
+ It would have been nice if the loss functions $\mathcal L_S$ and ${\mathcal L}_{KD} $ could have been defined.
+ There is no description of how the data was divided between training and validation.
+ What were the $\gamma_1^{(l)}$ and $\gamma_2^{(l)}$ set to in the experiments?

+ There was no analysis of the experimental improvements.  Are the improvements statistically significant?


**Strength And Weaknesses:**

The paper talks in generalities and employs a general abstract notation. There is no experimental specificity.


**Summary Of The Paper:**

The authors argue that contributions of knowledge from the teacher to the student network should be layer dependent.  Adaptive block-wise learning automatically balances the contribution of knowledge between the student and teacher for each block.


**Summary Of The Review:**

The paper is incrementally novel, but it has extensive experiments.
The paper talks in generalities without any technical or experimental details.  A below average ML paper.

---

### Official Review · Reviewer_BeZK · 2022-10-31

**Confidence:** 5
**Correctness:** 3
**Technical Novelty And Significance:** 2
**Empirical Novelty And Significance:** 2
**Recommendation:** 6

**Clarity, Quality, Novelty And Reproducibility:**

Clarity: good
Quality: good
Novelty: good

**Strength And Weaknesses:**

Strength:
+ Paper is well written
+ Experiments on common benchmark datasets for KD
+ A number of KD schemes are tested with the proposed scheme in the experimental results

Weaknesses:
1. The problem might not be well justified. The main argument of the paper is to use the performance of the proposed scheme to claim the solving of the fixed contributions of the two types of knowledge in different blocks. It would be more interesting to see what part of the network goes wrong with the fixed contributions scheme.

2. Whether the improvement of the network comes from the auxiliary network? Please consider adding an experiment by removing the bi-level optimization but still keeping all the auxiliary networks. Consider tuning the fixed parameters between ground truth knowledge and teacher knowledge in this scheme.

3. The increase in performance might be the result of solving the gradient vanishing not solving the balance between ground truth information and the teacher at the different blocks. The addition of an auxiliary could also be interpreted as a shortcut to the last layer. Consider checking the gradient vanishing between the standard KD scheme and the proposed KD scheme. Or further, can we just add a shortcut from each layer to the final layer and have different losses corresponding to each shortcut?

4. For some datasets, the utilization of teacher knowledge only could also achieve comparable or even better results compared to the utilization of both teacher knowledge and ground truth knowledge. The authors should consider adding a KD scheme with only teacher knowledge guidance as one of the baselines. So that the scheme of balancing between teacher knowledge and ground truth knowledge would be more meaningful.

5. How does the proposed scheme training time increase compared to the standard KD schemes? The main concern about the bi-level optimization is that it always takes too much time, while the achieved improvement in the manuscript seems to be not too significant (only around 0.5%, and poorer in some cases). This might be the strongest challenge to apply the proposed scheme to practical applications.

6. In table 3, Is that normal where the standard KD performs poorer than supervised training?

7. The authors should details out how each parameter is fixed in standard KD schemes in all experiments.

**Summary Of The Paper:**

The manuscript observes the problem of fixed contributions of ground truth knowledge and teacher knowledge at different blocks of the student networks during knowledge distillation training. The author proposes a bi-level optimization scheme to balance the knowledge on the lower level and update the network based on optimized balance at the higher level.

**Summary Of The Review:**

The paper might be helpful to the research community. However, the concern about training time might reduce the chances of its practical applications. While considering about the fundamental contributions, the authors should justify the problem clearly as well as put more analyses on the proposed schemes.

---

### Decision · Program_Chairs · 2023-01-20

**Decision:**

Reject

**Justification For Why Not Higher Score:**

The authors provided an extensive rebuttal however the two most important concerns have not been addressed adequately. In particular, there has been a questioning about the significance of the results in conjunction with the experimental setup, and this has not been alleviated while, at the same time, the code has not been provided.

**Justification For Why Not Lower Score:**

N/A

**Metareview: Summary, Strengths And Weaknesses:**


In this paper the authors question the hypothesis that knowledge from teacher and student contributes to the same extent to each student layer during knowledge distillation training. Therefore, they design a block-wise learning mechanism which balances the contribution of knowledge between student and teacher separately for each block.

Overall, this paper introduces an important finding: the unbalanced contributions of knowledge from the student and teacher per block. Furthermore, the authors design a scheme that uses this finding to improve training through balancing.

However, there are two key weaknesses: firstly, there are a quite a few "moving parts" in the proposed architecture and it is currently not entirely clear how these contribute to the performance of the method and generally the support of the motivation. In particular reviewer BeZK has asked a lot of clarification questions which have been to an extent answered in the rebuttal, however all this discussion is entirely ablation-driven; instead, it would be beneficial to have a more thorough theoretical analysis and relate more deeply to prior work like Neyshabur et al. 2020 mentioned by reviewer XFGM. A more fundamental question about the design of auxiliary classifiers (which affects greatly performance and efficiency) was posed by reviewer Tp5S but has not been addressed adequately. Again, a more theoretical driven complementary analysis (even if light) would have helped to guide the readers' intuition. I also sympathize with reviewer Tp5S who asks for comparison with other methods using auxiliary classifiers. I don't think this has to be an extensive comparison but it would be great to have an idea of how the common components behave in these methods, since these methods do have these common components even if they are motivated differently.

The second key weakness has been raised by reviewer Tp5S who points out that the proposed method has been used in combination with the baselines but without ensuring a common training setting. Given that the deltas are small, it makes one question whether this setup is convincing enough: could a slightly different setup move the deltas towards the other direction? This is important to understand especially since the authors provide only standard deviation of the results but they haven't performed a statistical test as reviewer Jk2C asked (this would be essential anyway because the deltas do seem small). Notice that this is an issue with convincingness of experiments and code is not provided by the authors.